# Low-Temperature and Low-Pressure Silicon Nitride Deposition by ECR-PECVD for Optical Waveguides

Dawson B. Bonneville *, Jeremy W. Miller, Caitlin Smyth, Peter Mascher and Jonathan D. B. Bradley

Department of Engineering Physics and Centre for Emerging Device Technologies, McMaster University, 1280 Main Street West, Hamilton, ON L8S 4L7, Canada; millej29@mcmaster.ca (J.W.M.); caitlin.smyth@outlook.com (C.S.); mascher@mcmaster.ca (P.M.); jbradley@mcmaster.ca (J.D.B.B.)
* Correspondence: bonnevd@mcmaster.ca

**Abstract:** We report on low-temperature and low-pressure deposition conditions of 140 °C and 1.5 mTorr, respectively, to achieve high-optical quality silicon nitride thin films. We deposit the silicon nitride films using an electron cyclotron resonance plasma-enhanced chemical vapour deposition (ECR-PECVD) chamber with Ar-diluted $SiH_4$, and $N_2$ gas. Variable-angle spectroscopic ellipsometry was used to determine the thickness and refractive index of the silicon nitride films, which ranged from 300 to 650 nm and 1.8 to 2.1 at 638 nm, respectively. We used Rutherford backscattering spectrometry to determine the chemical composition of the films, including oxygen contamination, and elastic recoil detection to characterize the removal of hydrogen after annealing. The as-deposited films are found to have variable relative silicon and nitrogen compositions with significant oxygen content and hydrogen incorporation of 10–20 and 17–21%, respectively. Atomic force microscopy measurements show a decrease in root mean square roughness after annealing for a variety of films. Prism coupling measurements show losses as low as 1.3, 0.3 and $1.5 \pm 0.1$ dB/cm at 638, 980 and 1550 nm, respectively, without the need for post-process annealing. Based on this study, we find that the as-deposited ECR-PECVD $SiO_xN_y$:$H_z$ films have a suitable thickness, refractive index and optical loss for their use in visible and near-infrared integrated photonic devices.

**Keywords:** Chemical vapour deposition Silicon nitride; Optical waveguides

## 1. Introduction

As thin-film silicon nitride ($Si_3N_4$) becomes a standard material in integrated photonics, the need for a broad range of deposition recipes for different applications becomes apparent. $Si_3N_4$ has many demonstrations as a low-loss waveguide material [1–6] for diverse applications due to its transparency in the visible and near-infrared (NIR), such as biological sensing and particle manipulation, environmental monitoring, optical interconnects and anti-reflective (AR) coatings [7–12]. Due to the need for $Si_3N_4$ integration with metal components or organic materials for many of these uses, it is advantageous to have flexible fabrication techniques that operate at lower deposition temperatures and pressures.

There is an extensive background of published literature on the formation of amorphous silicon nitride films through chemical vapour deposition (CVD) processes. Low-pressure CVD (LPCVD) is commonly used to achieve low-loss stoichiometric $Si_3N_4$ [13], features high deposition temperatures around 650 to 900 °C and has been incorporated in silicon photonics platforms [9,14,15]. However, these high deposition temperatures might pose fabrication constraints on silicon photonic chips with dopants and metal layers, or other substrates which are sensitive to higher temperatures. Most metals used in tunable heaters and optoelectronic devices can be damaged at temperatures exceeding ~400 °C, which makes post-metalized chips incompatible with LPCVD chambers. Plasma-enhanced CVD (PECVD) is also extensively used in the fabrication of silicon nitride thin films. PECVD systems operate below 450 °C, owing to the plasma operating as an auxiliary energy source,

and are more suited to post-fabrication processing. Low-loss silicon nitride waveguides have been fabricated using PECVD [16]. One main disadvantage of PECVD techniques is the relatively high concentration of incorporated hydrogen in the samples, resulting in low-density structures and producing low-refractive index films. These highly hydrogenated films also exhibit more significant optical losses in the C-band due to Si-H and N-H bonds [2]. It is possible to remove the hydrogen from the films either through annealing or utilizing precursor gases that do not contain hydrogen, such as switching from $NH_3$ to $N_2$ [17]. Other varieties of CVD techniques can help mitigate the high concentrations of hydrogen. These include inductively coupled plasma (ICP-PECVD) [18,19], high-density plasma (HDP-CVD) [13] and electron cyclotron resonance (ECR-PECVD) [20–24]. These techniques allow for high power densities, resulting in greater dissociation of the precursor gases and lower hydrogen incorporation into the films. Low pressures of 1–2 mTorr are realized in ECR-PECVD by using a strong magnet to confine and stabilize the plasma during deposition, whereas inductively coupled plasma (ICP) techniques typically operate at pressures >10 mTorr [13]. Interestingly, in the case of ICP-PECVD and ECR-PECVD, the reaction chamber is separate from the plasma chamber, allowing for high-quality films to be produced at low temperatures and pressures without the risk of surface damage from high energy bombardment. Limiting surface damage is of particular interest in waveguide fabrication as scattering at the material interface is a significant source of optical loss.

Optical waveguide fabrication through ECR-PECVD has been explored [16,25,26] in various systems, demonstrating it as a viable candidate for the fabrication of visible and NIR waveguides. Variation in film stoichiometry by changing the nitrogen/silane ($N_2$/$SiH_4$) gas ratio during ECR-PECVD fabrication has been investigated, with optical losses of <1.0 and 1.5 dB/cm at 1310 and 1550 nm for N-rich and nearly stoichiometric films, respectively [2]. However, films in [2] were deposited at 350 °C and a minimum chamber pressure of 650 mTorr, which is two orders of magnitude higher than the deposition pressures shown here. By measuring optical loss at visible and NIR telecom wavelengths, and characterizing film stoichiometry as well as unwanted contaminants such as oxygen, we demonstrate a non-stoichiometric film capable of suitable device performance, fabricated at relatively low deposition pressures and temperatures compared to the existing literature.

We report on hydrogenated silicon nitride films deposited using a low-temperature and low-pressure ECR-PECVD system. The stoichiometry of the films and the impacts of annealing on the hydrogen content are studied through Rutherford backscattering spectrometry (RBS) and elastic recoil detection analysis (ERDA), respectively. Low-pressure recipes ranging from 1.1 to 2.4 mTorr with deposition temperatures of 140 °C show pre-annealing optical losses in the visible to NIR range suitable for integrated optical devices. We demonstrate that the associated optical losses of hydrogenated non-stoichiometric PECVD silicon nitride ($SiO_xN_y$:$H_z$), as reported in the literature [13], are reasonable for specific applications requiring low-temperature deposition conditions and access to thicker films up to 650 nm. We characterized the thin film optical loss across a broad wavelength range including visible and telecommunication bands from 638 to 1640 nm using prism coupling and observed low surface roughness measured through atomic force microscopy (AFM). Low losses are achieved without post-fabrication annealing treatment, which is typically required to reduce significantly higher losses in the telecommunication bands due to hydrogen bonds [27–30]. These characteristics demonstrate ECR-PECVD to be a highly promising method for planar waveguide fabrication.

## 2. Fabrication

### 2.1. Silicon Nitride Thin-Film Deposition

We deposited silicon nitride thin films on various substrates using an ECR-PECVD system. The system is divided into three parts, allowing for a separate load-lock, main chamber and ECR plasma generation sections. The remotely generated ECR plasma allows for a high fractional ion density at low process pressure, limiting the damage to the film during growth. A forward microwave power of 500 W is applied through a quartz window

near an 875 G electromagnet with $N_2$, $O_2$ and Ar inlets for ECR plasma generation during deposition. In the ECR plasma generation section of the chamber, the strong electromagnet is used to minimize ionized plasma species sidewall reactions and confine electron motion to field lines concentrated towards the sample surface inside the main chamber. Due to the relatively small main chamber size of roughly 10 cm radius, deposition pressures in the 1–2 mTorr range are possible using ECR-assisted confinement. A more detailed description of the system and the orientation of the gas shower heads, ECR plasma section and sample holder can be found in Figure 22 and Figure 27 in [31]. These features are aimed at lowering the incorporation of interstitial hydrogen, an unwanted contamination which leads to strong absorption in the S and C telecommunication wavelength windows [2]. The system's low deposition pressure with operating conditions shown here of 1–2 mTorr also ensures a low concentration of Si-H bonds to form during deposition while using $SiH_4$ precursors, limiting interstitial hydrogen incorporation during deposition, as is a common drawback in PECVD systems. Another expected source of hydrogen is invasive $H_2O$ in the form of water vapour likely entering the chamber through the ~5 m-long gas transit lines and potentially residual from the sample transfer process. This also explains the presence of oxygen in the films which has been quantified with RBS. The deposition system uses dedicated turbopumps for both the main chamber and the separate load-lock chamber, keeping the main chamber isolated during venting for transfer. The system is also equipped with in situ sputtering [32] and metal organic delivery lines which provide in situ incorporation of rare earths for the development of solid-state lighting devices. Additionally, it has been used for the formation of silicon nanoclusters embedded in $Si_xN_y$ [33].

We deposited $SiO_xN_y$:$H_z$ films on different substrate materials mounted to a stainless-steel stage with tungsten clips to secure the samples. The sample stage sits inside the main chamber, mounted on a motor-controlled arm for rotation, holding the sample face normal to the precursor gas showerhead, and ECR-generated plasma. A heating element in contact with the back of the sample stage allows for evenly distributed heating, which provides additional surface energy for the reaction. As mentioned, the strong ECR plasma acts as an auxiliary power source for the reaction, allowing the lower deposition temperatures when compared to ICP depositions. Sample stage rotation was maintained at a constant 0.5 Hz during all of the depositions and the substrates were heated to 140 °C. The partial pressure from each introduced gas species was allowed to vary throughout the deposition whilst maintaining a constant gas flow rate. This resulted in the partial pressure from each gas being on average 5% higher by the end of the deposition. The partial pressures imparted by each of the gases at the start of depositions are shown in Table 1, with an assumed associated error of $\pm0.01$. Across the film recipes, $SiH_4$/Ar flow was held constant while the $N_2$/Ar flow was varied. This was due to the higher range of gas flow choices associated with higher flow rates from the mass flow controller (MFC) with $N_2$/Ar gas flow. Any change in the silane flow would result in a drastic stoichiometry variation even across 0.1 sccm of gas flow change. With an expected uncertainty and occasional variation of ~0.1 sccm of flow from any of the MFC units, a safer, smaller stoichiometry change is available by changing $N_2$/Ar flow across the ranges shown in Table 1. Silane and nitrogen dilution were 30/70% and 10/90% with argon, respectively. The sample stage is 3 inches in diameter and was used to fix multiple substrates to the stage at once. One round of thick (~300–450 nm) films were deposited on $2 \times 2$ cm$^2$ silicon pieces and on 6 μm of thermally grown $SiO_2$ on $5 \times 3$ cm$^2$ silicon pieces ($SiO_2$-Si) cleaved from a fully oxidized wafer, for ellipsometry and thin film optical loss measurements, respectively. The deposition time for the first round was 3.5 h, and 1 h for the second round of depositions, producing thinner films (~100 nm) for RBS and ERDA measurements on $2 \times 2$ cm$^2$ pieces of vitreous carbon and silicon, respectively. After preliminary studies on optical loss and recipe optimization had been performed, a final thicker (650 nm) film was deposited on silicon and a $5 \times 3$ cm$^2$ cleaved piece of an $SiO_2$-Si wafer to increase modal confinement in the $SiO_xN_y$:$H_z$ layer when performing optical loss measurements. The deposition parameters used for the

different film recipes are summarized in Table 1, including pre-deposition pressures, and pressures during depositions imparted by each partial gas flow.

**Table 1.** $SiO_xN_y:H_z$ thin-film deposition parameters.

| Deposition Parameters | Film 1 | Film 2 | Film 3 | Film 4 | Film 5 | Film 6 |
|---|---|---|---|---|---|---|
| $N_2/Ar$ (10/90%) flow rate (sccm) | 10.0 | 12.5 | 15.0 | 17.5 | 20.0 | 15.0 |
| $SiH_4/Ar$ (30/70%) flow rate (sccm) | 5.0 | 5.0 | 5.0 | 5.0 | 5.0 | 5.0 |
| $N_2/Ar$ partial pressure—round 1(mTorr) | 1.15 | 1.40 | 1.64 | 1.83 | 2.10 | 1.63 |
| $N_2/Ar$ partial pressure—round 2 (mTorr) | 1.18 | 1.43 | 1.62 | 1.87 | 2.09 | N/A |
| $SiH_4/Ar$ partial pressure—round 1 (mTorr) | 1.60 | 1.79 | 2.01 | 2.24 | 2.45 | 2.02 |
| $SiH_4/Ar$ partial pressure—round 2 (mTorr) | 1.62 | 1.81 | 1.98 | 2.24 | 2.49 | N/A |
| Pre-deposition pressure—round 1 (Torr) | $9.5 \times 10^{-8}$ | $8.0 \times 10^{-8}$ | $4.8 \times 10^{-8}$ | $5.6 \times 10^{-8}$ | $1.6 \times 10^{-7}$ | $7.5 \times 10^{-7}$ |
| Pre-deposition pressure—round 2 (Torr) | $2.2 \times 10^{-8}$ | $1.1 \times 10^{-7}$ | $7.5 \times 10^{-8}$ | $7.0 \times 10^{-8}$ | $8.5 \times 10^{-8}$ | N/A |
| Deposition time for round 1 (min) | 210 | 210 | 210 | 210 | 210 | 380 |
| Deposition time for round 2 (min) | 90 | 90 | 90 | 90 | 90 | N/A |
| Temperature (°C) | 140 | 140 | 140 | 140 | 140 | 140 |

## 2.2. Annealing

To investigate the presence and removal of hydrogen in the films, annealing under ambient $N_2$ flow was carried out in a quartz tube furnace. We selected 1000 °C for 1 h for hydrogen removal based on multiple findings [29,30]. Characteristics reported for hydrogen release include lower optical losses, decreased thicknesses and increased indices of refraction due to film densification. The films were annealed with an open exhaust and $N_2$ flow rate of 500 sccm in a 3-inch × 2-meter-long furnace. Films were inserted into the furnace at maximum temperature on a quartz wafer boat and pulled out of the furnace but left in the tube and allowed to cool adjacent to the heating zones during a 4 h ramp down.

## 3. Characterization and Results

### 3.1. Refractive Index and Thickness

We used a combination of variable-angle spectroscopic ellipsometry (VASE) and prism coupling to determine the refractive index and thickness of the $SiO_xN_y:H_z$ films on the silicon and thermally oxidized silicon pieces, respectively. The results of the refractive index and thickness measurements are displayed in Figure 1. The anticipated relationship was observed between the $N_2/Ar$ flow rate and refractive index, with films having a higher refractive index at decreased $N_2/Ar:SiH_4/Ar$ flow rate ratios due to increasing silicon content. An expected trend is observed in the deposition rate which increases from 1.43 to 2.11 nm/min for Film 1 and Film 5, respectively. This is correlated to the increase in $N_2/Ar$ flow rate providing a higher concentration of $N_2$ in the chamber and increased pressures during film growth. The films show the expected behaviour of decreased thickness after annealing, which can be attributed to hydrogen removal and film densification. Additionally, an increased refractive index is observed after annealing, which is consistent with film densification as has been seen in [28].

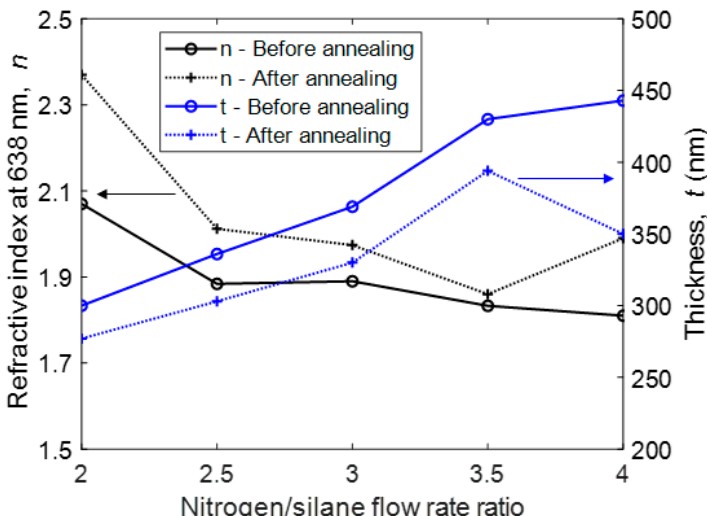

**Figure 1.** Refractive index and thickness of films deposited for 3.5 h at 140 °C with varying $N_2$/Ar:$SiH_4$/Ar flow rate ratios before and after annealing at 1000 °C for 1 h in ambient nitrogen (500 sccm). Connecting lines are to help guide the eye.

The films were grown from 300 to 450 nm thick to ensure a planar waveguide mode was supported at 1550 nm, without the need for a top cladding layer. This allowed for the refractive index, thickness and loss to be measured using prism coupling at longer wavelengths, including 1550 and 1640 nm. However, at these thicknesses, we observed the annealed films lifting off the edges of 80% of the samples which were deposited on large $SiO_2$-Si substrates, which is due to the known tensile stress in $Si_xN_y$ films. We measured film refractive indices in the range of 1.81–2.07 before annealing across the $N_2$/Ar:$SiH_4$/Ar flow ratios shown in Figure 1. Film 5 with the highest $N_2$/Ar flow showed a drastic thickness decrease of ~100 nm after annealing, while Film 1 with the lowest $N_2$/Ar flow showed a drastic increase in the index of ~0.33. This is anticipated to be a cause of Film 1 and Film 5 being, respectively, overly silicon- and nitrogen-rich and therefore in different dominant regimes for Si-H and N-H bonding. Film 5, moreover, is likely in a regime beyond the stoichiometric film ratio [2], causing a difference in bond reorganization in comparison to the other films after annealing.

### 3.2. Optical Loss

In order to study the $SiO_xN_y$:$H_z$ films as candidates for optical waveguides, a significant merit is a low optical loss. To quantify this, we used prism coupling to excite the transverse-electric (TE) polarized fundamental planar modes in the film and measure their decay as they propagate through the $SiO_xN_y$:$H_z$ layer with a scanning fiber normal to the surface and coupled to a photodetector. This was carried out before annealing at four different wavelengths including 638, 847, 1310 and 1550 nm, as shown in Figure 2.

The loss for Film 1 at 638 and 847 nm was too high to be measured on the system. This is anticipated because of the higher Si content in Film 1 as a result of a decreased $N_2$/Ar:$SiH_4$/Ar flow rate ratio. A similar trend is observed in Film 2, which exhibits high visible loss but promising losses in the 1310–1550 nm range. Films 3 and 4 demonstrate relatively low losses at all wavelengths and less than 0.5 dB/cm at 847 and 1310 nm, while Film 5 exhibits a higher loss at these wavelengths. Film 5 was deposited with the highest $N_2$/Ar:$SiH_4$/Ar flow rate ratio and the highest pre-deposition pressure. It is likely that this recipe leads to films rich in N-H bonds which is known to be an unwanted source of optical loss [2]. The increased loss at shorter wavelengths is anticipated to be a result of increased Si-H bonds in this deposition regime, potentially past the stoichiometric point of gas ratios.

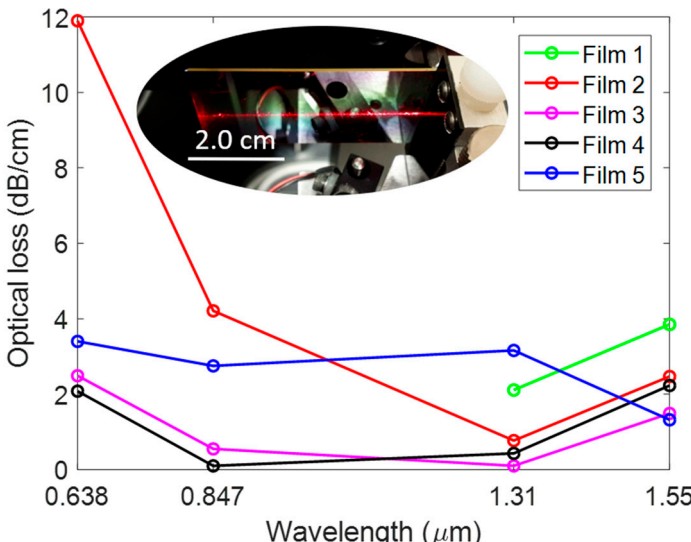

**Figure 2.** Optical loss determined via prism coupling at various wavelengths in the visible to infrared for each $SiO_xN_y:H_z$ film. The films were deposited on 6 µm thermally grown $SiO_2$ layers on Si substrates. Connecting lines are to help guide the eye. Inset: red light streak from Film 3 visible during 638 nm loss measurement.

In order to further investigate the films' optical quality, we measured the loss at a finer spacing of wavelengths from 1510 to 1600 nm, as shown in Figure 3a. The incorporation of hydrogen in the films is expected to lead to additional absorption in this wavelength regime [2]. Films 1–3 demonstrate higher loss around 1510 nm and a decrease in loss towards longer wavelengths past ~1540 nm, indicative of the presence of hydrogen in the films. Films 4 and 5, however, show consistent or increasing losses across this wavelength regime, indicative of other dominant loss mechanisms rather than hydrogen absorption, such as substrate interaction or intrinsic material loss. Film 2 was the only film which was measurable for loss after annealing, demonstrated as Film 2A in Figure 2a, showing a loss decrease from 1520 to 1550 nm. This was due to all other films deposited on $SiO_2$-Si lifting off or cracking during the annealing step. To increase modal interaction with the planar waveguide layer, we also deposited a thicker film with the same recipe as Film 3, shown in Table 1 denoted as Film 6. This recipe was chosen as it demonstrated the lowest losses on average from visible to NIR telecommunication wavelengths. The loss results for this 650 nm-thick film are shown in Figure 3b from 638 to 1620 nm with finer 1480–1580 nm spacing shown in the inset.

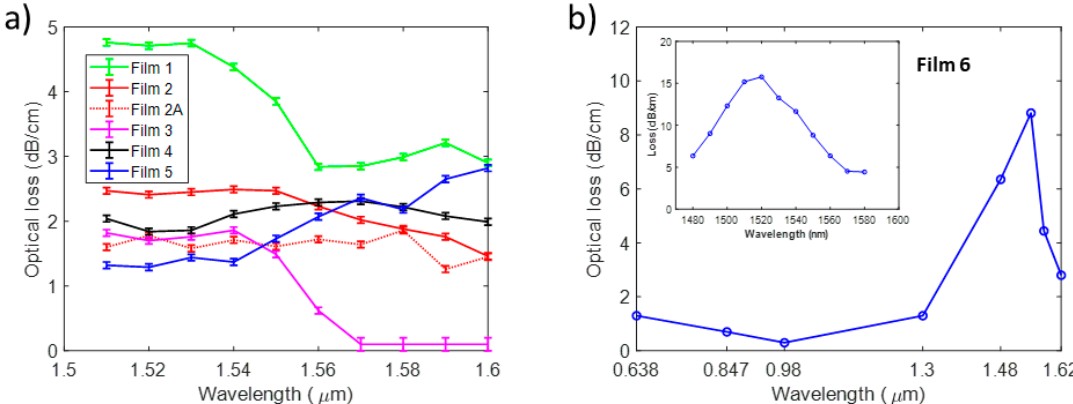

**Figure 3.** (**a**) Optical loss of various films determined via prism coupling from 1510 to 1600 nm. (**b**) Optical loss determined via prism coupling at various wavelengths in the visible to infrared for Film 6 and finer spaced loss measurements from 1480 to 1580 nm (inset). Connecting lines are to help guide the eye.

Each data point represents a loss measurement derived from a fit to an exponential loss curve obtained from a normally scanning fiber during measurement. The inherent uncertainty in this method is reflected in the data and increased for Film 3, which shows propagation losses on the order of 0.1 dB/cm beyond 1570 nm in Figure 3a, which is the measurable limit for the system. This loss trend is promising for applications in the telecommunications L-band (1570–1620 nm) and potentially longer wavelengths where new telecommunication and sensing windows are demanding low-loss waveguides. Figure 3b shows that the thicker film has significantly higher losses from 1510 to 1580 nm, with still relatively low losses at 638, 847, 980 and 1310 nm. Increasing the film thickness from 369 to 650 nm increases the modal interaction and overlap with the planar waveguide layer from ~70.0 to ~89.0 % at 1510 nm, while decreasing the interaction with the air from ~8.9 to ~3.5 %. Doing so resulted in a film which demonstrates visible–NIR losses as low as 0.3 dB/cm at 980 nm, with increased C-band losses as high as ~8.0 dB/cm at 1550 nm shown in Figure 3b and the inset, respectively. The increase in loss for Film 6 is expected to be attributed to the increased modal interaction with the hydrogen in the $SiO_xN_y:H_z$ films due to increased thickness. It may also be the case that increased pre-deposition pressures lead to higher hydrogen incorporation in comparison to Film 3.

Annealing the samples at 1000 °C for 1 h caused the majority of the films to be immeasurable due to stress-induced lifting of the film off the $SiO_2$, or in one case of Film 5, total failure showing visible cracks throughout the whole deposited layer. This was expected, as the films are relatively thick for $Si_xN_y$ depositions and therefore are expected to have high tensile stress. However, this was not the case for Film 2, which demonstrated a loss decrease of $0.8 \pm 0.1$ dB/cm from 1520 to 1550 nm after annealing and not lifting off. This might be attributed to the higher Si content in the film and decreased tensile stress. These data are shown as Film 2A in Figure 2a, with a dotted line to guide the eye, in comparison to the losses measured prior to annealing. The remaining ~1.6 dB/cm of loss in this range suggests that other loss mechanisms rather than hydrogen content still need to be overcome for the S–L bands in our ECR-PEVCD $SiO_xN_y:H_z$ films.

### 3.3. Surface Roughness

To quantify the surface roughness of the films, we used AFM to inspect $1 \times 1$ μm$^2$ areas on multiple locations for selected samples on Si substrates. Films 2 and 3 were chosen due to their drastically different optical losses for visible wavelengths. It is anticipated any surface roughness will increase scattering losses, especially in the visible regime with the known $1/\lambda^4$ relationship. Surface homogeneity was also inspected and used to quantify uncertainty on the root mean square (RMS) reported roughness and peak–peak amplitude values. A correlation between annealing and decreased peak–peak height is observed when measuring Film 3 before and after annealing; however, it is noted that for films of similar RMS roughness, drastically different (~10 dB/cm at 638 nm) optical loss is reported. A decrease in the peak–peak amplitude of $1.65 \pm 0.40$ nm is observed after annealing Film 3, which is attributed to the densification of the film. After annealing out the hydrogen in Film 2, it demonstrates a higher optical loss at 1550 nm compared to the unannealed Film 3, while still demonstrating less surface roughness (RMS and peak–peak) as shown in Table 2 which summarizes the AFM amplitude plots in Figure 4. This points to stoichiometry-dominated optical loss mechanisms for the ECR-PECVD $SiO_xN_y:H_z$ films. This can also be attributed to the low percentage of optical intensity in the upper air cladding, on the order of ~3–9% for most films measured. Moving forward, this is an important merit for multilayer systems, where interface losses may be exacerbated by higher surface roughness.

**Table 2.** Surface roughness from atomic force microscopy.

| Film # | 2, Annealed | 3, As-Deposited | 3, Annealed |
|---|---|---|---|
| RMS roughness (nm) | $0.95 \pm 0.01$ | $1.10 \pm 0.01$ | $1.01 \pm 0.05$ |
| Peak–peak amplitude (nm) | $9.55 \pm 0.16$ | $9.58 \pm 0.12$ | $7.93 \pm 0.38$ |

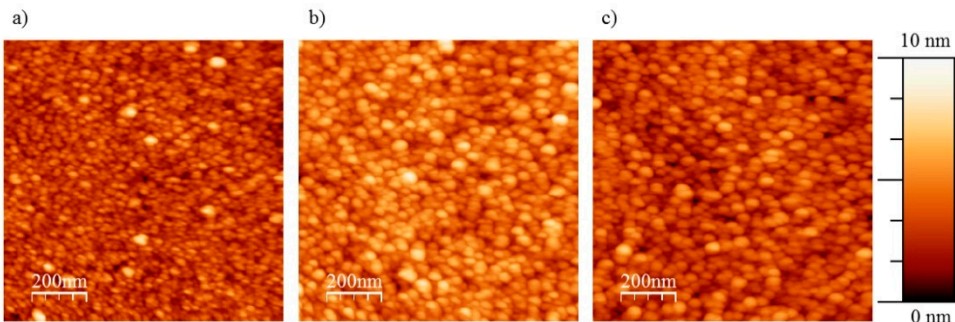

**Figure 4.** (**a**) AFM amplitude plot for Film 2 annealed, (**b**) Film 3 unannealed and (**c**) Film 3 annealed.

### 3.4. Stoichiometry

As shown in Table 1, a second round of depositions was performed to produce thinner films, on carbon substrates for RBS measurements, as well as $2 \times 2$ cm$^2$ Si pieces for annealing and ERDA measurements. The SiO$_x$N$_y$:H$_z$ films deposited on carbon substrates were used to measure RBS data while allowing for a separation of recoil energies between the film and substrate, while the films deposited on Si pieces were used to quantify the amount of hydrogen incorporation via ERDA. Data were collected using a Tandetron accelerator with He$^{4+}$ ions at a 1.8 MeV 4° incident for RBS, and a 2.9 MeV incident at a grazing angle of 75° for ERDA. The data obtained for each film are shown in Figure 5. The uncertainty is reflected in the data point size for RBS, while the ERDA data are fit with higher sample-dependent uncertainty due to a higher sensitivity with changing fit parameters.

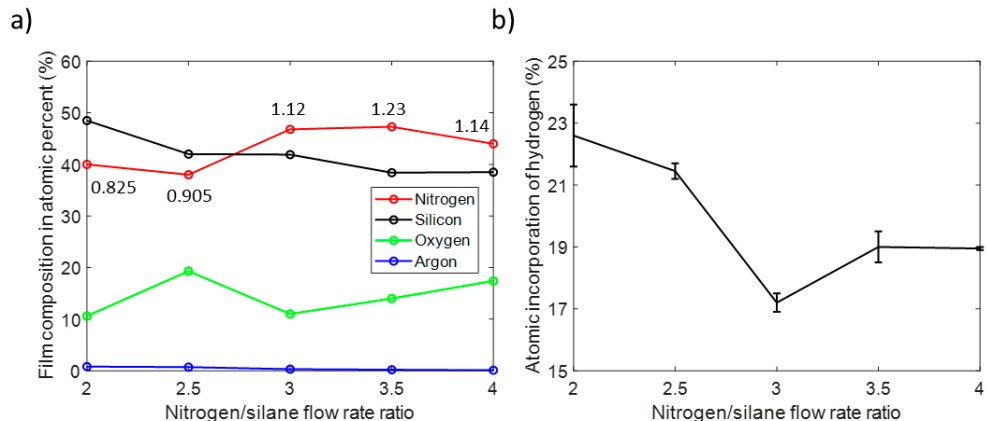

**Figure 5.** (**a**) Film composition determined from RBS excluding hydrogen with N/Si ratio indicated. (**b**) Hydrogen incorporation determined from ERDA before annealing. The incorporated hydrogen after annealing was measured to be below 0.1 at.% for all films. Connecting lines are to help guide the eye.

The deposition recipes studied produce films of varying N/Si composition ratios, which demonstrate oxygen and argon contents of 10–20 and < 1%, respectively. We observe the expected trend of decreasing Si content as the N$_2$/Ar flow increases, and conversely, the film composition becomes closer to a stoichiometric N/Si ratio of 1.333, as indicated in Figure 5a. However, this trend is no longer observed with further increasing N$_2$/Ar:SiH$_4$/Ar flow rate ratios, as shown in [2]. We observe less nitrogen content in Film 5, at 20 sccm of N$_2$/Ar flow, during deposition. This is also coupled with an increase in oxygen content in the film. The films contain 10 to 20% oxygen, as shown in Figure 5a, which would indicate the deposition of silicon-oxynitride films. However, the relatively high oxygen content does not correlate with the expected refractive index in such films, which typically have indices ranging from 1.48 to 1.68 using PECVD [34,35]. Future work aims to understand the distribution and bonding nature of O$_2$ in this system using Fourier

transform infrared spectroscopy (FTIR). The contamination is expected to be due to water vapour present in the vacuum chamber due to sample transfer, and possible $N_2$/Ar contamination with ambient $O_2$ from the gas transit lines before the MFC. The oxygen content is also anticipated to be a cause of the decreased N content as more Si-O bonds start to be favoured during deposition, rather than Si-N bond formation [36]. The abundance of oxygen in our film matrix can be explained by the presence of oxygen radicals in the plasma. At low $N_2$/Ar flow rates, there are low quantities of N radicals leading to Si-H or Si-Si bonds becoming favourable, resulting in the production of a silicon-rich film [37,38]. This effect is observed in Figure 5a, with a flow rate of 10 sccm of $N_2$/Ar producing a silicon-rich $SiO_xN_y$:$H_z$ film. ERD measurements show typical hydrogen incorporation of roughly 20% across the deposited films for ECR-PECVD deposition [13]. The lowest-loss film (Film 3) is also the film showing the lowest hydrogen incorporation of $17.2 \pm 0.5$ at.%. After annealing, all films show below 0.1 at.% hydrogen incorporation.

Future work will focus on minimizing hydrogen in the chamber by decreasing the $SiH_4$/Ar flow, and possibly including argon flow to keep operating pressures at ~1–2 mTorr while also decreasing the $N_2$/Ar flow. This may also be beneficial in reducing the $O_2$ content in the film or potentially providing insight into its sources. Additionally, longer pump-down times after sample transfer before deposition may achieve a lower concentration of water vapour in the chamber, as well as a chamber bake step, which will both be considered in the future. The recipes used have shown that a non-stoichiometric and medium hydrogenated film leads to thicknesses and optical losses acceptable for integrated photonic devices for the visible and NIR wavelength regions. Some recipes show promising losses in the important S, C and L communications bands even in unannealed films, and further investigation of deposition conditions and recipes can lead to reduced hydrogen incorporation and losses in that spectral region.

## 4. Conclusion

We deposited $SiO_xN_y$:$H_z$ films on various substrates using ECR-PECVD with a fixed $SiH_4$/Ar precursor and varied $N_2$/Ar flow at 140 °C from 1.60 to 2.45 mTorr. To characterize the films, we used VASE and prism coupling to measure the refractive index and thickness, as well as the optical loss, respectively, before and after annealing. Refractive indices ranging from 1.81 to 2.07 were measured before annealing with optical losses as low as 1.3, 0.3 and $1.5 \pm 0.1$ dB/cm at 638, 980 and 1550 nm, respectively, for films varying in thickness from 300 to 650 nm. Annealing for 1 h at 1000 °C in $N_2$ resulted in decreased thicknesses and increased refractive indices, pointing towards film densification, and hydrogen removal. Post-annealing loss measurements show a decrease of $0.8 \pm 0.1$ dB/cm from 1.52 to 1.57 µm. We quantified surface roughness with AFM, showing an RMS roughness of $1.0 \pm 0.1$ nm for unannealed and annealed films of varying optical losses. Annealing was shown to decrease the peak–peak roughness amplitude by $1.65 \pm 0.40$ nm. We verified hydrogen removal using ERDA and observed it to decrease from 17.2 to 22.6 to < 0.1 at.% after annealing. RBS was used to investigate the atomic composition of the films and showed films of varying N/Si stoichiometry with high oxygen content ranging from 10.6% to 19.3%. Despite the films being non-stoichiometric $SiO_xN_y$:$H_z$ and having medium hydrogen incorporation, low optical losses were still achieved from the visible to L telecommunication band using low deposition pressures and temperatures. This opens the possibility of applying such films in integrated photonic devices, particularly for post-processing fabrication of such devices on samples already containing temperature- or pressure-sensitive organic or metal layers. Future work aims to reduce the hydrogen incorporation in the films by reducing the $SiH_4$/Ar gas flow rate during deposition, to reduce and gain further insight into oxygen sources and bonding mechanisms in the films via FTIR and to characterize the stress in deposited $SiO_xN_y$:$H_z$ films.

**Author Contributions:** Conceptualization, J.D.B.B., P.M. and D.B.B.; Methodology, D.B.B., J.W.M. and C.S.; Software, D.B.B. and C.S.; Validation, D.B.B., J.D.B.B. and P.M.; Formal Analysis, D.B.B., J.W.M. and C.S.; Investigation, D.B.B., J.W.M. and C.S.; Resources, J.D.B.B., and P.M.; Data Curation, D.B.B., and J.W.M.; Writing—Draft Preparation, D.B.B.; Writing—Review & Editing, D.B.B., J.W.M., J.D.B.B. and P.M.; Visualization, D.B.B. and J.D.B.B.; Supervision, J.D.B.B.; Project Administration, J.D.B.B. and P.M.; Funding Acquisition, J.D.B.B. and P.M. All authors have read and agreed to the published version of the manuscript.

**Funding:** Natural Sciences and Engineering Research Council of Canada (NSERC) (STPGP 494306, RGPIN-2017-06423); Canadian Foundation for Innovation (CFI) (CFI Project # 35548).

**Institutional Review Board Statement:** Not applicable.

**Informed Consent Statement:** Not applicable.

**Data Availability Statement:** Data available in a publicly accessible repository.

**Acknowledgments:** We thank Doris Stevanovic and Shahram Tavakoli of the Centre for Emerging Device Technologies (CEDT) at McMaster University for their assistance with fabrication, and Lyudmila Goncharova and Jack Hendriks from Western University at the Tandetron facilities for assistance with RBS and EDRA.

**Conflicts of Interest:** The authors declare no conflict of interest.

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
