# Peer review of "Low-Temperature and Low-Pressure Silicon Nitride Deposition by ECR-PECVD for Optical Waveguides"

_applsci, doi:10.3390/app11052110_

Round 1
Reviewer 1 Report
Manuscript Number: applsci-1120540
Title: Low Temperature and Low Pressure Silicon Nitride Deposition by ECR-PECVD for Optical Waveguides
The authors present some results about the synthesis and characterization of Silicon Nitride thin films by ECR-PECVD. The paper is well written. I have some remarks and suggestions for the improvement of the paper.
Consequently, I suggest minor revision.
Detailed remarks are listed below:
- “he ECR plasma is generated at a distance from the sample holder to avoid ion bombardment of the thin film”. Could you precise this “distance”? Is it only supposed large enough? or experimentaly proved? Or numerically?
- “Due to the proximity of the SiH4 precursor showerhead to the sample surface, and the low pressure conditions, the mean-free path of the ionized species involved in the reaction can reach lengths comparable to the chamber dimensions [32].” It is hard to follow you without a sketch of the system. 1 to 2 mtorr correspond (very roughly) to 3 to 6 cm for the mfp. I am not sure that the chamber is so small, if I list all the components. I did not found in Ref 32 the explanation.
Second question about this sentence. What is the link between the mfp and the amount of Hydrogen incorporated? It does not seems so obvious. - “Sample stage rotation”. Same problematic. It is hard to understand the deposition chamber. How is placed the axis of the rotation compare to the vapor flow? What is the size of the stage. How many substrate can you fix on it?
- About the substrates… what mean “various”? Does it correspond to the number of substrates? Their size? Their nature (Si, Si+SiO2, carbon)? All of these points?
- I do not know if it is of your responsibility… but the reading of table 1 could be improved. For example by increasing the “position parameter” cell width and decreasing the “Film x” ones. Eventually a new column could be added with the “Round”
- Minor remark: why do you sometime use a dash between value and unit? (ex 4-hour)
- Figure 2, Inset. To what correspond the 1.5 cm readable on the wafer? Is it one of your samples? I did not find a mention of a full wafer as substrate. Could you precise this point?
- Rougness: what do you mean with the “peak-peak height”? Is it the maximum value of the height distribution of each picture or the average on 10 point (on each figure…)? More generally, I do not think it is so easy to discuss about so low roughness values. The reported error is less than an angstrom. Here, I completely agree with your assumption that stoichiometry is the dominant mechanism. In addition, unfortunately only 3 measurements are presented. I do not ask you to remove the discussion about the influence of the roughness but maybe this part could be improved.
- It is your choice, and it can be understand, but maybe, I would have appreciated the composition part not at the end. Eventually, it could help the discussion about the optical properties if you begin with this part.
Reviewer 2 Report
The manuscript presents the original research describing the low pressure and low temperature method to achieving the high optical quality silicon nitride thin films. In fact, the authors describe the resulting films as films of silicon oxynitride with a sufficiently high oxygen content. So, on my opinion, the authors must describe the sources of the oxygen introduction in the plasma. Moreover, it need to indicate the residual pressure in the vacuum chamber before deposition. Also, resulting films have the following designation in the literature: SiOxNy:H2.
